# Associative Conversation Model: Generating Visual Information from Textual Information

## Abstract

In this paper, we propose the Associative Conversation Model that generates visual information from textual information and uses it for generating sentences in order to utilize visual information in a dialogue system without image input. In research on Neural Machine Translation, there are studies that generate translated sentences using both images and sentences, and these studies show that visual information improves translation performance. However, it is not possible to use sentence generation algorithms using images for the dialogue systems since many text-based dialogue systems only accept text input. Our approach generates (*associates*) visual information from input text and generates response text using context vector fusing associative visual information and sentence textual information. A comparative experiment between our proposed model and a model without association showed that our proposed model is generating useful sentences by associating visual information related to sentences. Furthermore, analysis experiment of visual association showed that our proposed model generates (*associates*) visual information effective for sentence generation.

## 1 Introduction

As a model that can extract knowledge from conversations, the encoder-decoder model has been proposed (Sutskever et al., 2014) (Vinyals & Le, 2015). It consists of an encoder that encodes the input information into a context vector and a decoder that generates sentences using the context. Vinyals & Le (2015) showed that it is possible to extract knowledge and to conduct conversation by learning pairs of dialogues with the model. For example, Vinyals & Le (2015) reported that when asked who is Skywalker, their conversation model (*NCM*) responded "*he is a hero.*"

NCM has a problem that it is not possible to respond properly to the input texts that require visual information. For example, Vinyals & Le (2015) reported that when asked how many legs a spider have, NCM responded "*three, i think.*" Further, the image or video may contain more detailed information than texts. Consider, for example, a scene in a news program including a closed caption "*one marathon runner won the marathon competition*" and showing an image with the marathon runner with the gold medal. Here, in the video, more detailed information such as the gold medal that does not exist directly in the text is presented. We thought that if such detailed visual information could be extracted from the image, more specific and useful texts could be generated, including "*gold medals*" which can not be obtained with text alone. In recent years, studies have been reported in which translated sentences are generated by adding image features to the context vector encoded by the encoder-decoder model (Calixto et al., 2017) (Elliott & Kádár, 2017) (Nakayama & Nishida, 2017) (Saha et al., 2016) (Toyama et al., 2016). These studies showed that visual information works effectively for generating translation.

Meanwhile, visual information is not considered in many text-based dialogue systems, because what is given to the input is only the utterance text. How can the visual information be used without accepting visual information as the input to the dialogue system?

Based on the discussion above, we propose an Associative Conversation Model that associates the input text with the visual information and generates the response using both the text and the asso-

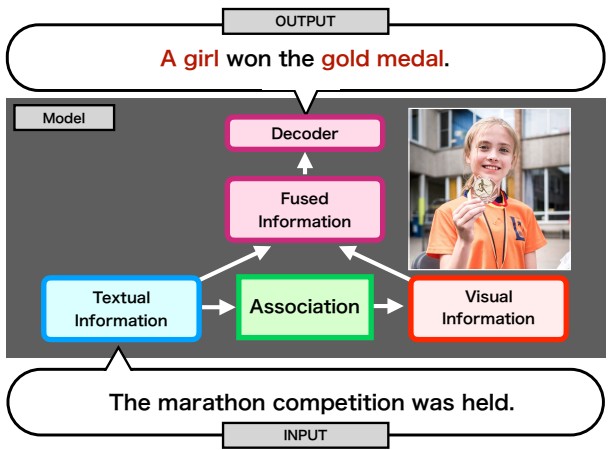

Figure 1: Generating a response by visual association. The textual information is used to estimate the corresponding visual information, and a response text is generated using the vector obtained by fusing the textual and visual information.

ciated visual information. In our proposed method, we attempted to generate response texts using visual information without inputting images. The contribution of this research is as follows:

- We made it possible to generate visual information related to sentence textual information through end-to-end learning of dialogue.
- We made it possible to generate sentences using visual information without directly inputting visual information by association.
- Our proposed model can generate response texts including useful information compared with a model without association by associating visual information related to input text.
  Our method is useful for constructing the text-based dialogue systems that automatically extract information from the text and the video data (e.g., TV news) to generate sentences.

## 2 ASSOCIATIVE CONVERSATION MODEL

### 2.1 OVERVIEW OF ARCHITECTURE

Figure 1 shows the overview of our generation-based model. The objective of this model is to enable for the decoder to generate a more appropriate response text which cannot be obtained only by textual information, by adaptively referring to the context vector based on visual information. However, in this task, only the textual description is given as input. Therefore, we adopted the following approach so that visual information can be utilized from the textual information.

- First, visual association is performed from the input text, and the visual information corresponding to the text is generated.
  (For example, figure 1 shows that the input text "*The marathon competition was held*" is used to generate the visual information corresponding to a scene where a girl runner in marathon competition won the gold medal.)
- Next, by fusing the textual information and the associated visual information is obtained the information reflecting either or both as necessary.
  (Figure 1 shows that the fused information is generated from the textual information "*marathon competition*" and the visual information "*gold medal*", "*girl*".)
- Finally, a response text is generated based on the fused information.
  (Figure 1 shows that the fused information was used to decode the final response "*A girl won the gold medal*".)

Our approach is simple. When generating sentences from the input texts and videos, an encoder for text and another for video can be used to acquire context vectors for text and for video, respec-

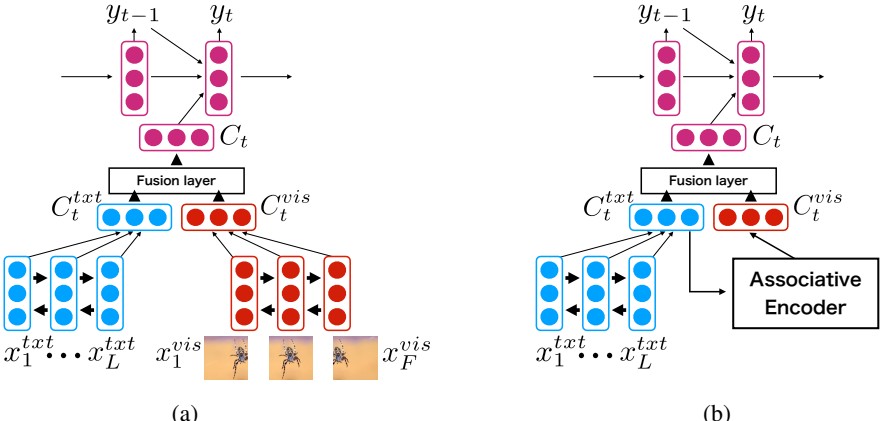

(a)  (b)

Figure 2: **(a)** Step1 : A model that performs prior learning for extracting context vectors $C_{txt}$ and $C_{vis}$. After this model learns $C_{txt}$ and $C_{vis}$ from video $X_{vis} = (x_1^{vis}, ..., x_F^{vis})$ and sentence $X_{txt} = (x_1^{txt}, ..., x_L^{txt})$ , we extract $C_{txt}$ and $C_{vis}$ using this model. **(b)** Step3 : The Associative Conversation Model. This model learns response texts using the associative visual information instead of visual information directly obtained from video.

tively. In our task, however, the videos cannot be directly obtained from the input because only the textual information is given as input. Therefore, our idea is to replace the encoder for video with a mechanism for generating the visual context vectors from texts. The configuration of this associative conversation model is simple, and is composed of the encoder for text and the decoder equipped with attention mechanism, as well as the visual association encoder from texts. Associative encoder is an RNN which generates visual context vectors from textual context vectors (see Sec 2.2.2). We used LSTMs for the textual encoder, the decoder, and the associative encoder, respectively (Hochreiter & Schmidhuber, 1997). The input to the associative conversation model is textual descriptions only, and the output is a response text to the input. This model generates the response text by inputting to the decoder the fused context vector obtained by fusing the textual context vector and the visual context vector generated from the input text. The associative conversation model is able to learn the pairs of dialogue sentences by end-to-end, like a normal encoder-decoder model. Actually, however, two prior learnings are performed before training the associative conversation model. One is the training of multimodal encoder-decoder model which generates a response text from the video and text. The other is the training of the associative encoder which predicts the visual context vector from the textual context vector obtained by the multimodal encoder-decoder model.

## 2.2 LEARNING METHOD

Learning consists of the following three steps. **In order to learn the Associative Conversation Model in the final step 3, it is necessary to learn two models in step 1 and step 2, in advance.**

**Step 1: Extraction of context vectors between textual and visual information**

Figure 2a shows a network model used in step 1. End-to-end learning is performed on the model that inputs the utterance text $X_{txt}$ and the video $X_{vis}$ corresponding to the utterance and outputs the response sentence $Y$. In this learning, the following four components are trained simultaneously.

- Textual encoder inputting $X_{txt}$ and outputting $C_{txt}$ which is the textual context vector
- Visual encoder inputting $X_{vis}$ and outputting $C_{vis}$ which is the visual context vector
- Fusion layer inputting $C_{txt}$ and $C_{vis}$ and outputting the fused context vector $C$
- Decoder with attention mechanism, inputting $C$ and outputting the response text $Y$

The trained model is used to extract the correspondence between the textual and visual information (see Fig. 2a).

**Step 2: Learning for visual association**

In this step, the associative encoder is trained, which inputs the textual context vector $C_{txt}$ extracted in step 1 and outputs the visual context vector $C_{vis}$ corresponding to the input utterance text.

**Step 3: Generation of response text via association**

In step 3, learning is performed in the network where the visual encoder in step 1 is replaced with the associative encoder trained in step 2. This model is a generation-based model which inputs $X_{txt}$, and outputs a response text $Y$, by using the fused context vector $C$ obtained from $C_{txt}$ and $C_{vis}$ in the fusion layer. That is, the structure of the Associative Conversation Model is the same as the network of step 1, except that it uses an associative encoder instead of the visual encoder. In the fusion layer, the decoder with attention is trained again after the weights learned in step 1 are initialized. In textual and video encoders, the weights trained in step 1 are left unchanged, and are not updated. It should be noted that what is trained in this model are only the decoder, the attention, and the fusion layer.

### 2.2.1 STEP 1: EXTRACTION OF CONTEXT VECTORS BETWEEN TEXTUAL AND VISUAL INFORMATION

The training data used in this step are the utterance text, the video corresponding to the utterance text, and the response text for the utterance. We used videos rather than images because the texts often include the expression of actions. We created the dataset based on TV programs as they contain the utterance texts, their corresponding videos, and their response texts. First, the closed caption texts were extracted from TV news programs as the utterance texts $X_{txt}$. Also, the scenes corresponding to the temporal intervals where the utterances occurred were extracted as the sequence of images $X_{vis}$ from the TV news programs. For more information on the dataset, see Sec 3. Here, the utterance $X_{txt}$ is represented by the sequence of the corresponding word $x^{txt}$. Similarly, the video $X_{vis}$ corresponding to the utterance text is represented by the sequence of image features $x^{vis}$. In this work, the image features acquired by the already learned CNN were used instead of learning the image features directly from the input video (We used VGG16 (Simonyan & Zisserman, 2014)). Therefore, the input to the prior learning models are the sequence of word vectors $X_{txt} = (x_1^{txt}, ..., x_L^{txt})$ and the sequence of image feature vectors $X_{vis} = (x_1^{vis}, ..., x_F^{vis})$. Also, the output is the sequence of words vectors $Y = (y_1, ..., y_T)$. Here, $L$, $F$, and $T$ represent the length of each input text, the number of the input images, and the length of the output text, respectively. The model used in step 1 is the multimodal encoder-decoder model consisting of a textual encoder, a visual encoder, and a decoder with attention (Bahdanau et al., 2014) . In step 1, the textual and visual encoders encode the input text and video, respectively, to obtain the context vectors $C_t^{txt}$ and $C_t^{vis}$. Then, $C_t$ which is the fused context vector is input to the decoder to generate a response text. For extracting context vector $C_t$, the attention mechanism was used.

$$s_t = LSTM(C_t, s_{t-1}, y_{t-1}) \tag{1}$$

$$P(y|s_{t-1}, C_t) = softmax(W_s s_{t-1} + W_c C_t + b_s) \tag{2}$$

In the equations above, $s_t$ is the hidden state of the Decoder when generating the $t$ th output word. $W_s$, $W_c$ and the bias $b_s$ are learnable parameters. Attention mechanism can extract a context vector that strongly reflects some parts of the sequence corresponding to the words of the response text. For example, when the word "*spider*" of the text and the image of the spider were paid attention to, the textual and visual context vectors of the spider are generated, respectively. In this case, there is a correspondence relations between the textual and visual context vectors. So, in step 2, the associative encoder which predicts the visual context vectors from the textual context vectors is trained. The purpose of step 1 is to extract the textual and visual context vectors having correspondence relations to be used for learning in step 2. The usual attention-based encoder-decoder model calculates the context vectors by Eq. (3), (4), and (5). $h_i$ is the intermediate vector of the bidirectional LSTM computed from the input sequence $h_i = [\overrightarrow{h_i}; \overleftarrow{h_i}]$ (Schuster & Paliwal, 1997).

$$C_t^{att} = \sum_{i=0}^{T} \alpha_{t,i} h_i \tag{3}$$

$$\alpha_{t,i} = softmax(e_{t,i}) \tag{4}$$

$$e_{t,i} = w_e^{\mathrm{T}} tanh(W_e s_{t-1} + V_e h_i + b_e) \tag{5}$$

In the equations above, $e_{t,i}$ is the associated energy between Encoder hidden state $h_i$ and Decoder hidden state $s_{t-1}$, and $\alpha_{t,i}$ is the probability of it. $w_e$, $W_e$, $V_e$ and the bias $b_e$ are learnable parameters. Meanwhile, the multimodal encoder-decoder model acquires two context vectors $C_t^{txt}$ and $C_t^{vis}$. Thus, the multimodal encoder-decoder model uses the following equations instead of Eq. (3) (4) and (5).

$$C_t^{txt} = \sum_{i=0}^{L} \alpha_{t,i}^{txt} h_i^{txt} \tag{6}$$

$$C_t^{vis} = \sum_{j=0}^{F} \alpha_{t,j}^{vis} h_j^{vis} \tag{7}$$

$$\alpha_{t,i}^{txt} = softmax(e_{t,i}^{txt}) \tag{8}$$

$$\alpha_{t,j}^{vis} = softmax(e_{t,j}^{vis}) \tag{9}$$

$$e_{t,i}^{txt} = w_{e_{txt}}^{\mathrm{T}} tanh(W_{e_{txt}} s_{t-1} + V_{e_{txt}} h_i^{txt} + b_{e_{txt}}) \tag{10}$$

$$e_{t,j}^{vis} = w_{e_{vis}}^{\mathrm{T}} tanh(W_{e_{vis}} s_{t-1} + V_{e_{vis}} h_j^{vis} + b_{e_{vis}}) \tag{11}$$

Also, there are some responses that do not always require visual information in the dialogue. Therefore, the decoder needs to be able to measure which of the textual and visual information should be paid attention to and how strongly it should be focused. The degree to which the textual and visual context vectors $C_t^{txt}$ and $C_t^{vis}$ should be referred to can be learnt by introducing the weight matrices and by multiplying them to $C_t^{txt}$ and $C_t^{vis}$, respectively. Therefore, the context vector eventually passed to the decoder is the fused context vector having how strongly the textual and visual information should be referred to.

$$C_t = W_{txt} C_t^{txt} + W_{vis} C_t^{vis} + b_c \tag{12}$$

We learn the parameters $W_{txt}$, $W_{vis}$, and the biases $b_c$. This fused context vector $C_t$ is used to predict the next word in the decoder. We call this layer the fusion layer. In the step 1, the loss function for the associative encoder is given by:

$$L(\Theta, \Omega, K, \Lambda, H) = -\sum_{i=1}^{B} \sum_{j=1}^{T} y_{i,j} \log \hat{y}_{i,j} \tag{13}$$

where $\Theta$ denotes the parameters of the Decoder, and $\Omega$ denotes the parameters of the fusion layer, $K$ denotes the parameters of Attention FNN (See Eq. (10), (11)), and $\Lambda$ denotes the parameters of the textual encoder, and $H$ denotes the parameters of the visual encoder, and $B$ is the total number of time-steps, and $\hat{y}_i$ is vector of class probabilities at time-step $i$, and $y_i$ is one hot label vector. After training the multi-modal encoder-decoder model, the training data is again input to the model to save the textual and visual context vectors. The teacher forcing was used when training this model and extracting context vectors.

### 2.2.2 STEP 2: LEARNING FOR VISUAL ASSOCIATION

In step 2, the visual associative encoder is trained. The associative encoder is trained so that it can predict the visual context vector from the textual context vector. The Recurrent Neural Network (RNN) is used for the associative encoder. Accordingly, the associative encoder can be seen as a regression model with the input sequence of context vectors of words $C^{txt} = (C_1^{txt}, ..., C_T^{txt})$ and the output sequence of context vectors of images $C^{vis} = (C_1^{vis}, ..., C_T^{vis})$ (Eq. (14)). Here, $T$ is the length of the output text.

$$\hat{C}_t^{vis} = RNN(C_t^{txt}) \tag{14}$$

In the equations above, $\hat{C}_t^{vis}$ is visual context vector predicted from textual context vector $C_t^{txt}$. The loss function for the associative encoder is given by:

$$L(\Gamma) = \sum_{i=0}^{T} (C_i^{vis} - RNN(C_i^{txt}))^2 \tag{15}$$

where $\Gamma$ denotes the parameters of the RNN. We used LSTM which is capable of learning long-term dependence as the RNN.

### 2.2.3 STEP 3: GENERATION OF RESPONSE TEXT VIA ASSOCIATION

In step 3, the associative conversation model learns response texts using the associative visual information instead of visual information directly obtained from video (see Fig. 2b). The input to the associative conversation model is the sequence of words $X_{txt} = (x_1^{txt}, ..., x_L^{txt})$. The output is also the sequence of words $Y = (y_1, ..., y_T)$.

In the associative conversation model, the visual encoder obtained in the prior learning of step 1 is replaced with the associative encoder. Therefore, the architecture of the associative conversation model can be obtained by replacing Eq. (7) in step 1 with Eq. (14). In other words, the mechanism of blending the textual and visual information trained in step 1 is left unchanged, and the visual context vector is predicted from the textual context vector by the associative encoder. The visual context vectors as well as the textual context vectors are given to the decoder, and the decoder, the attention, and the fusion layer (Eq. (12)) are trained. The weights of the decoder, the attention and the fusion layer are initialized, and the weights obtained by prior learning of step 1 is not used. The weights in the textual and associative encoders are not updated, and the same parameters used in the prior learning of step 1 are given. The reason for re-training the decoder, the attention and the fusion layer is because the associative visual context has different property from the visual context vector generated in step 1. The associative visual context is different from the visual context vector obtained in step 1. Therefore, in the step 3, the loss function for the associative encoder is given by:

$$L(\Theta, \Omega, K) = -\sum_{i=1}^{B}\sum_{j=1}^{T} y_{i,j} \log \hat{y}_{i,j} \tag{16}$$

We thought that re-training parts other than encoding of context vector enables to generate response texts using the associative visual elements in common knowledge.

## 3 DATASET

The model was learned using the following data collected independently: subtitles in TV drama and the corresponding video where the subtitles were displayed. The visual encoder of the model of step 1 encodes a sequence of images (video).

Note that the subtitles were delimited by "!" , "?", or Japanese period to acquire sentences, and a pair of one sentence and the following sentence was regarded as one dialogue. Therefore, the response text of a certain utterance becomes the next input utterance. Also, the video was cut out as a frame sequence with a frame rate of 5 fps. Each frame was input to the prior learned convolutional neural network, and the output of the last pooling layer was used as the image features. VGG16 was used for the convolutional neural network (Simonyan & Zisserman, 2014). The recorded programs were 163 Japanese TV news broadcast from December 2016 to March 2017, having 38K dialogues and 19K vocabulary words. Data was divided into 34K dialogue pairs for training data and 4K for test data.

## 4 EXPERIMENTS

In this section, the result of comparison is shown, between the Associative Conversation Model and a model without association. An encoder-decoder model with attention mechanism for generating dialogue responses were used as the baseline model. The baseline model generates a sentence $Y$ from the input utterance text $X_{txt}$. Both the proposed model and the baseline model were trained with the same dialogue sentences, and we investigated the effect of association. We also analyzed whether the associative conversation model acquired effective visual information for response generation by visualizing the associated objects. As a result of the experiments, we found that the association works effectively to generate sentences with useful information. We also found that our proposed model associates visual information related to input texts.

Table 1: The accuracy of the news test data

| Model | Accuracy (News test data) [%] |
|---|---|
| Seq2Seq | 8.228 |
| Seq2Seq-Assc | 8.109 |

Table 2: Results of Human Evaluation

| Model | News | | Dialog | |
|---|---|---|---|---|
| | Mean $Acc_{L2}$@1 | Mean $Acc_{L1,L2}$@1 | Mean $Acc_{L2}$@1 | Mean $Acc_{L1,L2}$@1 |
| Seq2Seq | 0.066 | 0.429 | **0.01** | **0.251** |
| Seq2Seq-Assc | **0.109** | **0.503** | 0.00 | 0.180 |

## 4.1 MODEL CONFIGURATION

In step 1, we used single layer LSTMs as both the encoders and the decoder. The hidden layer dimension of the textual encoder was set to 512, the hidden layer dimension of the visual encoder to 512, the dimension of the fused context vector to 1024, and the hidden layer dimension of the decoder to 1024, the dimensionality of word embedding to 512, the dimensionality of image feature to 512, and the batch size to 64. Adagrad was used for optimization (Duchi et al., 2011).

In step 2, the associative encoder is composed of 4-layer LSTM with the hidden layers dimension being 1024, and the batch size was set to 64. Adam was used for optimization (Kingma & Ba, 2014).

In step 3, the model is composed of the textual encoder, the decoder, and the associative encoder instead of the visual encoder. The hidden layer dimension of the textual encoder was set to 512, the dimension of the fused context vector to 1024, the hidden layer dimension of the decoder to 1024, the dimensionality of word embedding to 512, and the batch size to 64. Adagrad was used for optimization.

Also, in step 3, the same weights and biases of the decoder and attention as in step 1 were used and not updated. The weights of associative encoder were not updated either. The weights of the decoder were initialized in step 3 (See Eq. (1), (2), (10), (11), and (12)) .

The baseline model is composed of single layer LSTMs as the encoder and the decoder. The hidden layer dimension of the textual encoder was set to 512, the dimension of the fused context vector to 1024, the hidden layer dimension of the decoder to 1024, the dimensionality of word embedding to 512, and the batch size to 64. Adagrad was used for optimization.

## 4.2 QUANTITATIVE EVALUATION

In order to compare the effect of association with and without association, we presented sentences generated by our model (*Seq2Seq-Assc*) and the baseline model (*Seq2Seq*) to six different humans and made it judge the score for the input utterances. The accuracy of the news test data of these models was 8.109% for the proposed model (*Seq2Seq-Assc*) and 8.228% for the baseline model (*Seq2Seq*) (Tab. 1). As the evaluation method, the method in NTCIR 13 STC-2 Japanese subtask was used (Shang et al., 2017). For evaluation of the generated sentences, a score of any one of 0, 1 or 2 is given to the four criteria; fluency, coherence, context-dependence, and informativeness.

The labels L0, L1, and L2 are given by the procedure called Rule-1 in NTCIR 13 STC-2 Japanese subtask (Shang et al., 2017). The procedure is given in listing 1.

Listing 1: Rule-1

```
IF fluent & coherent = 1
        IF context-dependent & informative = 2
                THEN L2
        ELSE L1
ELSE
        L0
```

As with Shang et al. (2017), Accuracy $Acc_G@k$ was calculated based on the following equation.

$$Acc_G@k = \frac{1}{nk} \sum_{r=1}^{k} \sum_{i=1}^{n} \delta(l_i(r) \in G) \tag{17}$$

$l_i(r)$ is the label assigned to the $r$ th response candidate for the $i$ th utterance sentence. $n$ is the number of labels assigned by evaluators to one response ($n = 7$). $G$ is the set of labels regarded as "correct" ($G = \{L2\}$ or $G = \{L2, L1\}$). $k$ is the number of response candidate sentences per utterance. In this experiment, since the model generates one response per input utterance, $k = 1$. Therefore, $Acc_{L2}@1$ is the average number of $L2$ labels given to the first response. In this experiment, 50 utterances of news subtitles and dialogue sentences including questions that ask general facts (e.g., "*What is the hair color of Donald Trump?*") were used as the evaluation data.

Table 2 shows the evaluation result. Both the proposed model (*Seq2Seq-Assc*) and the baseline model (*Seq2Seq*) gave the higher accuracy for the news data than those for the dialogue data. This is considered to be due to the fact that it is difficult for both of the two models learned from the news sentences to generate a response that satisfies the fluency of the dialogue sentence, since the expressions used in the dialogue sentences differ from those in the news sentences. The proposed method showed higher accuracy for the news sentences than the conventional method. In Mean $Acc_{L2}@1$, the proposed method gave higher accuracy than the conventional method, which can be seen that more responses with usefulness and context dependency were generated by the proposed method. On the other hand, Mean $Acc_{L1,L2}@1$ of the baseline model was higher for the dialogue sentence, but Mean $Acc_{L2}@1$ was almost 0.0 for both models.

### 4.3 Qualitative Evaluation

Figure 3 shows the texts generated from the test data by our model or by the baseline, and the images that are similar to the associated visual information. The Japanese sentences generated by the models were translated into English. There results show that our model generated texts with more useful information than the baseline. For example, in the example on the left of the figure 3, the proposed model generated a specific weather forecast with the word "*snowy*" for the input sentence "*The University Entrance exam will be held on 14th and 15th.*" Note that it is a fact that the snow actually fell on the day of the exam, and that the images showing that it was snowing at the venue of the exam were included in the training data. The important point here is that the word "*snowy*" cannot be easily generated from the input sentences alone, but is a word that can be generated for the first time in association with the image of snow.

On the other hand, the result generated by the baseline is "*There will be a large-scale fire that is also in western Japan and eastern Japan*" and contains erroneous information such as "*fire*". In the example on the right of figure 3, the proposed model generated the text including the word "*gold medal*" for the input sentence "*Well, today is All Japan Figure Skating Championships.*" In addition to this example, it was confirmed that the proposed model with the associative function generated useful sentences with more specific information than the baseline without it. These results suggest that association works effectively to generate sentences with more specific information. In order to verify this, we analyzed what kind of visual information was generated by the associative encoder.

### 4.4 Analysis of Visual Association

Figure 3 shows the analysis result of the association. These results show that our model successfully associated visual information related to the input sentence. The upper image in figure 3 is the image with the highest similarity to the associative visual context generated by association from the input sentence by the proposed model. We calculated the cosine similarity between the visual context vector $C_{vis}$ obtained in step 1 and the associative visual context vector $C'_{vis}$ generated by the associative encoder. Also, we assumed that the images that were paid attention to at the time indicated by the value of $\alpha^{vis}$ which is derived when the most similar context vector $C_{vis}$ was generated, are associative images. In other words, the image in figure 3 was obtained by visualizing the associative

---

[1] Source: Image on the left: "NHK News 7" broadcast on NHK on 11th January 2017, Image on the right: "NHK News 7" broadcast on NHK on 23rd February 2017

| Input | The University Entrance exam will be held on 14th and 15th. | Well, today is All Japan Figure Skating Championships. |
|---|---|---|
| Output by Baseline | There will be a large-scale fire that is also in western Japan and eastern Japan. | Aiming for four consecutive championships in the women's singles, athletes of the Japanese championships participated in the tournament. |
| Output by ACM | It is highly expected to be **snowy** and windy. | A player who has won the **gold medal** in women's singles. |
| Image associated from the input | | |
| Words generated mainly from the associated image | **Snowy** | **Gold medal** |
| Cos similarity | **0.333** | **0.344** |

Figure 3: Example of comparison results on validity of sentence generation by visual association[1]. It indicates that the visual information associated from the input text by the proposed model exhibits some correspondence with the input text. In addition, it shows that the proposed model can generate response texts including useful information compared with the model without association. For example, the result on the left shows that our model associates the snow scene with the utterance text and generated the word "*snowy*". Note that it is a fact that it actually snowed on the day of the exam. The important point here is that the word "*snowy*" cannot be easily generated from the input sentences alone, but is a word that can be generated for the first time in association with the image of snow.

visual information from texts using images in the training data. From the result of figure 3, it is confirmed that the visual information matching the contents of the sentence can be associated. For example, the example to the right of figure 3 shows that a scene where a skating player acquired a gold medal was obtained by visual association on the topic of "*Skating Championships*", although it misunderstood figure skating as speed skating. Also, the word "*gold medal*" was generated from its association result. Figure 4 shows the attention weights when generating the word "*gold medal*". It does not pay attention to the word "*Figure*". However, It is shown that the word "*gold medal*" is generated paying attention to the words "*Skating*" and "*Championships*".

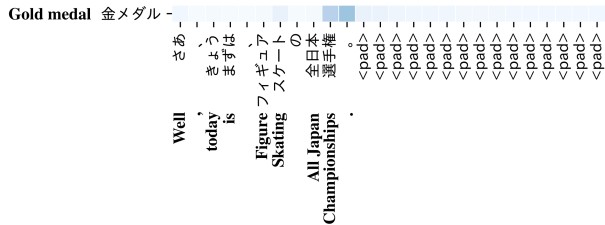

Figure 4: The heatmap that visualized attention weights when generating the word "gold medal"

The associated image includes a player holding the gold medal and other players skating behind her. Multiple examples were found, in which the images matching the input sentence were successfully associated like this. Some of these examples are found in the appendix. Also, these examples show that useful words were generated which are difficult to generate easily without any association. For example, it is difficult to generate the word "*snowy*" from the sentence "*The University Entrance*

*exam will be held on 14th and 15th*", without the visual association that it actually snowed around the venue at that time.

However, there remain several problems. In some examples, it was confirmed that the visual information was associated with a topic different from the input sentence. For example, in the example above, a scene where a speed skating player won the gold medal was associated with the topic about figure skating. It is expected that this can be improved by increasing the amount of the training data. Also, the proposed model could not generate a good reply to the utterances in a colloquial tone, because the learning was performed using the training data composed of sentences of news programs. A solution to this is to learn by training data composed of general dialogue sentences instead of news sentences when re-learning the model using the fused context vectors in step 3. That is, using the knowledge extracted by the encoders in steps 1 and 2, relearning is performed in step 3 using general dialogue sentences. Once the knowledge can be extracted, the model can be trained more efficiently by performing only step 3 according to the task (e.g., general conversation).

## 5 RELATED WORK

In studies about conversation model based on maltimedia data, Mostafazadeh et al. (2017) used deep neural network models trained on social media data (pair of image and text), and show their approach improves the quality of response generation. Although they assumed Image-Grounded Conversations, we developed their idea and assumed the case of not using the images during the conversation. In recent years, studies have been reported in which translated sentences are generated by adding image features to the context vector encoded by the encoder-decoder model (Calixto et al., 2017) (Elliott & Kádár, 2017) (Nakayama & Nishida, 2017) (Saha et al., 2016) (Toyama et al., 2016). These studies showed that visual information works effectively for generating translation. The approaches that use images only when training such as our approach are Toyama et al. (2016) and Elliott & Kádár (2017).

Toyama et al. (2016) proposed a neural machine translation model that introduces a continuous latent variable containing an underlying semantic extracted from texts and images. Their model is the encoder-decoder with attention, but there is a mechanism to generate the latent variables. When for predicting a translation in decoder inputting the sentence to encoder, decoder generate a target word using a latent variable generated from the source text, target text, and an image.

Elliott & Kádár (2017) show their approach using multitask learning model, inputting the text and outputting the text and the image feature, improves translation performance. The network of structure is attention-based encoder-decoder, but has two decoders outputting text or image feature vector $\hat{v}$. This decoder outputting image feature vector is trained to predict the true image vector $v$.

There are important differences between our approach and their approaches. First, although their approach has given the image as a visual information when training, our approach has given the video (sequence of image features) to expand their approach. By this extension, we think our model is able to learn words expressing motion effectively. Secondly, we are dealing the dialogue task, not the translation task. Thirdly, we challenged our model extract knowledge from the noisy data that the visual information at that time and the sentence does not necessarily correspond (News of the video and subtitle). Therefore, our model includes a fusion layer learning to use the measure of visual information and textual information (Eq. (12)). The difference between Elliott & Kádár (2017) and our approach is our approach is not multitask learning model, thus our model receives input of visual information. Hence it is possible to perform attention to visual information during training can be extracted visual information according to the response word.

## 6 CONCLUSIONS

In a study applying a sentence generation algorithm of translation sentence to a conversation model, there was a problem that it was not possible to respond well to an input text which requires visual information. However, it is not possible to use sentence generation algorithms using images for the dialogue systems since many text-based dialogue systems only accept text input. Based on the discussion above, we propose an Associative Conversation Model that associates the input text with the visual information and generates the response using both the text and the associated visual infor-

mation. Comparative experiments with models that do not use association show that association of visual information related to input texts produces response texts that contain valuable information compared to models without association. Analysis of association also showed that our proposed method can generate visual information related to sentence textual information through end-to-end learning of dialogue. Our method is useful for constructing the text-based dialogue systems that automatically extract information from the text and the video data (e.g., TV news) to generate sentences.

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

## 7 APPENDIX

Additional examples of comparison results on validity of sentence generation by visual association are shown in figure 5 and figure 6. In the example on the left of figure 5, "*Yokozuna*" which represents the highest rank in sumo, was visually associated from the input, but the name "*Hakuho*" who is also Yokozuna and is different from the person in the associated image, was output. In the example on the right of figure 5, a pitcher was visually associated, but he is a different person from "*Otani*" who is another pitcher described in the input. In the example on the left of figure 6, the proposed model associated an image showing the placement of high pressures from the input phrase "*high pressure*", and the word "*sunny*" was generated from the associated image. In the example on the right of figure 6, the proposed model associated an image of a person riding a bicycle likely to fall over due to a strong wind, from the input phrase of low pressure. The words "*traffic*" and "*windstorm*" were generated from the associated image

| Input | The Grand Sumo Tournament is in the second day. | As for pitchers, three players including Otani have been selected from Nippon Ham. |
|---|---|---|
| **Output by Baseline** | No. 1 is No. 1 in 1 meter. | Baseball is out in the professional this season. |
| **Output by ACM** | Today, **Yokozuna** Hakuho will aim for the first victory. | This is an **athlete**. |
| **Image associated from the input** |  |  |
| **Words generated mainly from the associated image** | **Yokozuna** (The highest rank in sumo) | **Athlete** |
| **Cos similarity** | **0.336** | **0.297** |

Figure 5: Topic : Sports [2]

| Input | The vicinity of Honshu is expected to be covered widely by mobile high pressure. | In tomorrow morning, it will be isolated snowstorms around the Japan Sea side of western Japan and eastern Japan, due to the developing low pressure. | |
|---|---|---|---|
| **Output by Baseline** | Let's move on to the weather around the country on tomorrow. | Vigilance is necessary for windstorms and high waves. | |
| **Output by ACM** | Tomorrow morning, it will be **sunny** in many places from western Japan to eastern Japan, and the side on the Japan Sea of western Japan. | Please also be aware of the influence on **traffic** caused by **windstorms**, heavy blizzards, and snowdrift. | |
| **Image associated from the input** |  |  | |
| **Words generated mainly from the associated image** | **Sunny** | **Traffic** | **Windstorms** |
| **Cos similarity** | **0.749** | **0.338** | **0.280** |

Figure 6: Topic : Weather [3]

---

[2] Source: Image on the left: "NHK News 7" broadcast on NHK on 13th January 2017, Image on the right: "News Watch 9" broadcast on NHK on 17th February 2017

[3] Source: Image on the left: "News Watch 9" broadcast on NHK on 27th February 2017, Image on the right: "NEWS CHECK 11" broadcast on NHK on 20th February 2017

