# OpenReview forum: "Associative Conversation Model: Generating Visual Information from Textual Information"
_ICLR.cc/2018/Conference — Reject_

### Official Review · AnonReviewer2 · 2017-11-24
**A deep learning model is proposed for dialogue systems that takes advantage of visual information. The proposed method is the straightforward implementation of known models, and the evaluation is purely qualitative in a non standard scenario.**

**Rating:** 4
**Confidence:** 5

**Review:**



The authors describe a method to be used in text dialogue systems. The contribution of the paper relies on the usage of visual information to enhance the performance of a dialogue system. An input phrase is expanded with visual information (visual context vectors), next visual and textual information is merged in a deep model that provides the final answer.

Although  the idea is interesting, I found the methodology fairly straightforward (only using known models in each step) and not having a particular component as contribution. What is more, the experimental evaluation is limited and purely qualitative: a few examples are provided. Also, I am not convinced on the evaluation protocol: the authors used captions from  video+videos as input data, and used "the next caption" as the response of the dialogue system. I am not very familiar with this type of systems, but it is clear to me that the evaluation is biased and does not prove the working hypothesis of the authors. Finally, no comparison with related works is provided.


English needs to be polished

---

### Official Review · AnonReviewer3 · 2017-11-28
**The paper proposes a novel dataset for grounded dialog and suggests that one should reason about vision even when performing text-based dialog. While the idea is intuitive the execution is currently lacking in multiple respects and would benefit from a major revision.**

**Rating:** 3
**Confidence:** 4

**Review:**

**Strengths**
In general, the paper makes an important observation that even in textual dialog, it might often make sense to reason or “imagine” how visual instances look, and this can lead to better more grounded dialog.

**Weakness**
In general, the paper has some major weaknesses in how the dataset has been constructed, details of the models provided and generally the novelty of the proposed model. While the model on its own is not very novel, the paper does make an interesting computational observation that it could help to reason about vision even in textual dialog, but the execution of the dataset curation is not satisfactory, making the computational contribution less interesting.

More specific details below:
1. The paper does not write down an objective that they are optimizing for any of the three stages in the model, and it is unclear what is the objective especially for the video context prediction task -- the distribution over the space of images (or videos) for a given piece of text is likely multimodal and gaussian likelihoods might not be sufficient to model this properly. Not clear if the sequence to sequence models are used in teacher forcing model when training in Stage 1, or there is sampling going on. In general, the paper lacks rigor in writing down what it optimizes, and laying out details of the model clearly.

2. The manner in which the dataset has been constructed is unsatisfying -- it assumes that two consecutive pieces of subtitles in news channels constitutes a dialog. This is very likely an incorrect and unsatisfying assumption which does not take into account narrative, context etc. Right now the dataset seems more like skip-thought vectors [A] which models the distribution over contextual sentences given a source sentence than any kind of dialog.

3. The setup and ultimately the motivation in context of the setup is fairly artificial -- the dataset does have images corresponding to each “dialog” so it is unclear why the associative model is needed in this case. Further, it would have been useful to see quantitative evaluation of the proposed approach or statistics of the dataset to establish context for the dataset being a valid benchmark, and providing a baseline / numerical checkpoint for future works to compare to. Without any of these things, the work seems fairly incomplete.

Clarity:
1. Figure 2 captions are pretty unclear and hard to understand what they are conveying.
2. For a large part the paper talks about how visual instances are not available for textual phrases and then proceeds to assume access to aligned text and visual data. It would be good to clarify from the start that the model does need paired videos and text, and state exactly how much aligned data is needed.
3. Already learned CNN (Page. 4, Sec. 2.2.1): Would be good to mention which CNN was used.
4. Page 4: “the textual and visual context vectors of the spider are generated, respectively”: Would be good to clarify that textual and visual context vectors for the spider are attended to, as opposed to saying they are generated.

References:

[A]: Kiros, Ryan, Yukun Zhu, Ruslan R. Salakhutdinov, Richard Zemel, Raquel Urtasun, Antonio Torralba, and Sanja Fidler. 2015. “Skip-Thought Vectors.” In Advances in Neural Information Processing Systems 28, edited by C. Cortes, N. D. Lawrence, D. D. Lee, M. Sugiyama, and R. Garnett, 3294–3302. Curran Associates, Inc.

---

### Official Review · AnonReviewer1 · 2017-12-01
**The paper's main idea is interesting, and fairly well-explained.  However, outside of a couple of simple, interesting examples, no real validation of the technique is done, and therefore its value is not clear.**

**Rating:** 3
**Confidence:** 5

**Review:**

The paper proposes to augment (traditional) text-based sentence generation/dialogue approaches by incorporating visual information.  The idea is that associating visual information with input text, and using that associated visual information as additional input will produce better output text than using only the original input text.

The basic idea is to collect a bunch of data consisting of both text and associated images or video.  Here, this was done using Japanese news programs.  The text+image/video is used to train a model that requires both as input and that encodes both as context vectors, which are then combined and decoded into output text.  Next, the image inputs are eliminated, with the encoded image context vector being instead associatively predicted directly from the encoded text context vector (why not also use the input text to help predict the visual context?), which is still obtained from the text input, as before.  The result is a model that can make use of the text-visual associations without needing visual stimuli.  This is a nice idea.

Actually, based on the brief discussion in Section 2.2.2, it occurs to me that the model  might not really be learning visual context vectors associatively, or, that this doesn't really have meaning in some sense.  Does it make sense to say that what it is really doing is just learning to associate other concepts/words with the input text, and that it is using the augmenting visual information in the training data to provide those associations?  Is this worth talking about?

Unfortunately, while the idea has merit, and I'd like to see it pursued, the paper suffers from a fatal lack of validation/evaluation, which is very curious, given the amount of data that was collected, the fact that the authors have both a training and a test set, and that there are several natural ways such an evaluation might be performed.  The two examples of Fig 3 and the additional four examples in the appendix are nice for demonstrating some specific successes or weaknesses of the model, but they are in no way sufficient for evaluation of the system, to demonstrate its accuracy or value in general.

Perhaps the most obvious thing that should be done is to report the model's accuracy for reproducing the news dialogue, that is, how accurately is the next sentence predicted by the baseline and ACM models over the training instances and over the test data?  How does this compare with other state-of-the-art models for dialogue generation trained on this data (perhaps trained only on the textual part of the data in some cases)?

Second, some measure of accuracy for recall of the associative image context vector should be reported; for example, on average, how close (cosine similarity or some other appropriate measure) is the associatively recalled image context vector to the target image context vector?  On average?  Best case?  Worst case?  How often is this associative vector closer to a confounding image vector than an appropriate one?

A third natural kind of validation would be some form of study employing human subjects to test it's quality as a generator of dialogue.

One thing to note, the example of learning to associate the snowy image with the text about university entrance exams demonstrates that the model is memorizing rather than generalizing.  In general, this is a false association (that is, in general, there is no reason that snow should be associated with exams on the 14th and 15th—the month is not mentioned, which might justify such an association.)

Another thought: did you try not retraining the decoder and attention mechanisms for step 3?  In theory, if step 2 is successful, the retraining should not be necessary.  To the extent that it is necessary, step 2 has failed to accurately predict visual context from text.  This seems like an interesting avenue to explore (and is obviously related to the second type of validation suggested above).  Also, in addition to the baseline model, it seems like it would be good to compare a model that uses actual visual input and the model of step 1 against the model of step 3 (possibly bot retrained and not retrained) to see the effect on the outputs generated—how well do each of these do at predicting the next sentence on both training and test sets?

Other concerns:

1. The paper is too long by almost a page in main content.

2. The paper exhibits significant English grammar and usage issues and should be carefully proofed by a native speaker.

3. There are lots of undefined variables in the Eqs. (s, W_s, W_c, b_s, e_t,i, etc.)  Given the context and associated discussion, it is almost possible to sort out what all of them mean, but brief careful definitions should be given for clarity.

4. Using news broadcasts as a substitute for true dialogue data seems kind of problematic, though I see why it was done.

---

### Author Response · Authors · 2018-01-06
**Comments on the revision of the paper**

Dear reviewers,

We really appreciate your constructive and helpful suggestions.
We attempted to address all the points raised by the reviewers as much as possible, and modified the following points.

1. We added the section on the experiment of subjective evaluation (Sec.4.2).
2. We added the description of what is being optimized in each step (Eq.13, 14, 15).
3. We added brief definitions of several undefined variables.
4. We added the results of calculating the cosine similarity between the visual context vector obtained in step 1 and the associative visual context vector generated by the associative encoder, to show how close the associatively recalled image context vector is to the target image context vector (Fig.3, 5, 6).
5. We added the accuracy of each model when the test data were used (Tab.1).
6. We clarified the teacher forcing was used when training the model in Step 1.
7. The caption in Figure 2 was described in more detail.
8. We added a heat map of Attention in Figure 4, in order to show how precisely the context vector is associated as visual information.

Besides these points above, we replaced the image used in Fig.1 with another due to the copyright.
Also, we added the source descriptions of some images obtained from TV programs.

The followings are our responses to the points raised by the reviewers.

-- The model might not really be learning visual context vectors associatively.
Although the model could not fully acquire the precise and useful concepts we aimed at the beginning, we think that the model has learned to associate texts with the concrete information which can be obtained for the first time by associating visual information, as shown in section 4.4.

-- Lack of amount of data
It was impossible to collect the large amounts of data due to the time constraint.

-- What is the reason for using the news data ?
The first reason is that there exists no appropriate corpus where the texts are provided in the form of dialogues and which are composed of the texts and the videos representing the content corresponding to the texts. The second reason is that the narration in the news programs could be considered to be in the form of dialogue when supposing each sentence be the utterance and the subsequent response in the dialogue. We, however, think this procedure is problematic and we will conduct another experiment using more appropriate data.

-- Why not also use the input text to help predict the visual context?
We considered the input texts were not necessary for predicting the visual context vector because the textual context vector should be able to reflect the information of the input text in itself.

-- It may be good not to re-learn the decoder of step 3 and the mechanism of attention.
Thank you for the valuable comment. We would definitely like to complete the corresponding experiment.

---

### Decision · Program_Chairs · 2018-01-29
**ICLR 2018 Conference Acceptance Decision**

**Decision:**

Reject

**Comment:**

None of the reviewers are enthusiastic about the paper, primarily due to lack of proper evaluation.  The response of the authors towards this criticism is also not sufficient.  The final results are mixed which does not show very clearly that the presented associative model performs better than the sole seq2seq baseline that the authors use for comparison.  We think that addressing these immediate concerns would improve the quality of this paper.